# High Detection Frequency of Vaccine-Associated Polioviruses and Non-Polio Enteroviruses in the Stools of Asymptomatic Infants from the Free State Province, South Africa

**DOI:** 10.3390/microorganisms12050920

**Published:** 2024-04-30

**Authors:** Milton T. Mogotsi, Ayodeji E. Ogunbayo, Hester G. O’Neill, Martin M. Nyaga

**Affiliations:** 1Next Generation Sequencing Unit and Division of Virology, Faculty of Health Sciences, University of the Free State, Bloemfontein 9300, South Africa; tmogotsi16@gmail.com (M.T.M.); emmanuelngs1@gmail.com (A.E.O.); 2Department of Microbiology and Biochemistry, Faculty of Natural and Agricultural Sciences, University of the Free State, Bloemfontein 9300, South Africa; oneillhg@ufs.ac.za

**Keywords:** enteroviruses, NPEV, poliovirus, echoviruses, coxsackievirus, infants, South Africa

## Abstract

Enterovirus (EV) infections are widespread and associated with a range of clinical conditions, from encephalitis to meningitis, gastroenteritis, and acute flaccid paralysis. Knowledge about the circulation of EVs in neonatal age and early infancy is scarce, especially in Africa. This study aimed to unveil the frequency and diversity of EVs circulating in apparently healthy newborns from the Free State Province, South Africa (SA). For this purpose, longitudinally collected faecal specimens (May 2021–February 2022) from a cohort of 17 asymptomatic infants were analysed using metagenomic next-generation sequencing. Overall, seven different non-polio EV (NPEV) subtypes belonging to EV-B and EV-C species were identified, while viruses classified under EV-A and EV-D species could not be characterised at the sub-species level. Additionally, under EV-C species, two vaccine-related poliovirus subtypes (PV1 and PV3) were identified. The most prevalent NPEV species was EV-B (16/17, 94.1%), followed by EV-A (3/17, 17.6%), and EV-D (4/17, 23.5%). Within EV-B, the commonly identified NPEV types included echoviruses 6, 13, 15, and 19 (E6, E13, E15, and E19), and coxsackievirus B2 (CVB2), whereas enterovirus C99 (EV-C99) and coxsackievirus A19 (CVA19) were the only two NPEVs identified under EV-C species. Sabin PV1 and PV3 strains were predominantly detected during the first week of birth and 6–8 week time points, respectively, corresponding with the OPV vaccination schedule in South Africa. A total of 11 complete/near-complete genomes were identified from seven NPEV subtypes, and phylogenetic analysis of the three EV-C99 identified revealed that our strains were closely related to other strains from Cameroon and Brazil, suggesting global distribution of these strains. This study provides an insight into the frequency and diversity of EVs circulating in asymptomatic infants from the Free State Province, with the predominance of subtypes from EV-B and EV-C species. This data will be helpful to researchers looking into strategies for the control and treatment of EV infection.

## 1. Introduction

Enteroviruses (EVs) are among the most common agents infecting humans worldwide, classified under the genus *Enterovirus* in the family *Picornaviridae* [1,2,3]. The EV capsid encloses a positive sense, single-stranded (+ss)RNA genome of approximately 7.5 kb in length, with a single open reading frame (ORF) encoding a polyprotein that is further cleaved to yield four structural proteins (VP1-VP4) and seven non-structural proteins (2A to 2C and 3A to 3D) [4]. The classification of EVs is largely based on the nucleotide (nt) sequence of the viral capsid protein VP1 coding region [5], and according to the latest International Committee on Classification of Viruses (ICTV) report, the genus *Enterovirus* comprises twelve EV species (EV-A to EV-L) and three human rhinovirus (HRV) species (HRV-A, HRV-B, and HRV-C), and over 300 serotypes have been identified [6]. EV-A to EV-D species predominantly infect humans, while other species infect a range of mammalian hosts, including livestock and non-human primates [5,7,8,9,10,11,12,13,14].

Infections due to EVs in newborns usually result in asymptomatic infections to severe, life-threatening diseases with manifestations such as aseptic meningitis, myocarditis, hepatitis, encephalitis, acute flaccid paralysis (AFP), and neonatal enteroviral sepsis [15,16,17]. Newborns are more vulnerable to EV infections, with an incidence of seven cases per 1000 newborns [18,19,20].

The enterovirus species B (EV-B) members include echoviruses and coxsackievirus B, which are the most common causes of EV infections in humans and responsible for the majority of the severe EV infections in neonates [19,21,22]. Poliovirus (PV), the most extensively studied subtype of EV-C species and the aetiologic agent of poliomyelitis, consists of three serotypes, i.e., PV1, PV2, and PV3 [23,24]. PV infection can be asymptomatic or may result in conditions such as AFP [3]. To date, humans remain the only known host and reservoir of PV, and it is the only EV for which a vaccine is available.

In 1988, the World Health Assembly (WHA) resolved to eradicate poliomyelitis by the year 2000, leading to the establishment of the Global Polio Eradication Initiative (GPEI) [25]. Prior to the polio eradication era, polio had high endemicity in many African countries, with over 1590 cases recorded on the continent in 1995 [26]. As a result of effective immunisation and intensive surveillance, by 2015, there were zero cases of wild-type polioviruses. Currently, wild poliovirus (WPV) type 1 remains endemic only in Afghanistan and Pakistan [27].

Vaccination has been the main approach to polio across the globe, and in South Africa, the primary vaccination schedule for poliovirus consisted of a trivalent live-attenuated oral polio vaccine (tOPV), which comprised all three serotypes 1, 2, and 3, and a trivalent inactivated polio vaccine (tIPV) [28]. However, in 2016, as part of the global eradication of poliovirus type 2, South Africa participated in the global switch from tOPV to bivalent oral polio vaccine (bOPV), comprising only type 1 and type 3 [29]. OPV is administered at birth and 6 weeks, and IPV is administered as part of a hexavalent vaccine at 6, 10, and 14 weeks [30].

By September 2019, PV3 was eradicated [30]; thus, PV1 is the only wild-type PV circulation. The last wild PV case in South Africa was reported in 1989 [31]. A recent study examined over two thousand cases of AFP in South Africa for the period from 2016 to 2019 and reported zero detection of wild-type polioviruses, with only Sabin vaccine strains type 1 or 3 detected in less than 1% of the samples tested [29].

Nevertheless, in South Africa, the focus has been more on AFP cases [29]. The current study, therefore, aimed to describe the prevalence and diversity of human enteroviruses in a cohort of asymptomatic infants from the Free State Province, South Africa, using viral metagenomics.

## 2. Materials and Methods

### 2.1. Ethical Approval

This study was conducted with the approval of the Free State Department of Health and the University of the Free State Health Sciences Research Ethics Committee (HSREC), with ethics number UFS-HSD2020/0327/2710. 

### 2.2. Participant Description and Sample Collection

In May 2021, newborns were recruited at three public hospitals around Mangaung in the Free State province of South Africa. Faecal samples were collected longitudinally at four-time intervals from the first week of birth until the age of six months. Demographic and clinical information were obtained along with sample collection. Faecal samples were sent, in an icebox, to the University of the Free State-Next Generation Sequencing (UFS-NGS) Unit, Bloemfontein, South Africa, for viral metagenomic analysis. 

### 2.3. Inclusion and Exclusion Criteria

The inclusion criteria considered newborns at the age of zero during recruitment, newborns who will be residing within the Mangaung metropolitan region during the study, and newborns with or without clinical symptoms. Regarding the exclusion criteria, the study excluded newborns with congenital abnormalities that required lengthy hospitalisation.

### 2.4. Sample Processing and Nucleic Acid Extraction

Stool samples were enriched for virus particles using the NetoVIR protocol [32], with modifications. Briefly, 10% faecal suspension was prepared in phosphate-buffered saline (PBS) (Sigma-Aldrich, St. Louis, MO, USA) and homogenised at 3000 rpm for 1 min using a Beadbug homogeniser (Benchmark Scientific, Sayreville, NJ, USA). The homogenate was clarified by centrifugation at 13500 rpm for 3 min, followed by filtration through a 0.45 µm filter (GVS, Bologna, Italy) to remove bacterial and eukaryotic cells. The filtrate was treated with benzonase (Merck, Burlington, MA, USA) and micrococcal nucleases (New England Biolabs, Ipswich, MA, USA) for 2 h at 37 °C to digest non-protected nucleic acids. Nuclease digestion was terminated with 0.5 M EDTA. Viral nucleic acid extraction was carried out using the QIAamp Viral RNA Mini kit (Qiagen, Hilden, Germany) without carrier RNA.

### 2.5. Reverse Transcription and Random Amplification

Complementary DNA and random amplification were performed with the QIAseq FX Single Cell RNA Library Preparation Kit (Qiagen, Hilden, Germany), followed by measurement of DNA concentrations on a Qubit 3.0 fluorometer using the Qubit dsDNA High Sensitivity Assay kit (Thermo Fischer Scientific, Waltham, MA, USA).

### 2.6. DNA Library Preparations and Next Generation Sequencing

DNA libraries were constructed from the amplified cDNA and uniquely barcoded using the QIAseq FX Single Cell RNA Library Preparation Kit (Qiagen, Hilden, Germany), following the manufacturer’s instructions. Purified and size selection was carried out by using Ampure XP beads (Beckman Coulter, Brea, CA, USA). Library fragments were assessed for quality on an Agilent 2100 Bioanalyzer (Agilent Technologies, Santa Clara, CA, USA) using the dsDNA High Sensitivity Assay kit (Agilent Technologies, Santa Clara, CA, USA). Final libraries were normalised, multiplexed, and sequenced on an Illumina MiSeq platform (Illumina, San Diego, CA, USA) using a v3 Reagent Kit to generate 2 × 150 bp paired-end reads.

### 2.7. Genome Assembly and Virus Identification

Metagenomic sequences generated from the Illumina MiSeq platform were analysed in Genome Detective [33], a streamlined automated web-based bioinformatic pipeline for sequence assembly and pathogen identification. Briefly, low-quality raw reads in fastq format were filtered and trimmed using Trimmomatic, and FastQC is used to visualise the quality of the processed reads. De novo assembly is performed with metaSPAdes, followed by virus identification and assignment of serotypes/genotypes using the enterovirus genotyping tool embedded in Genome Detective [33].

### 2.8. Phylogenetic Analysis

Multiple sequence alignments of our study strains and selected reference sequences were performed using the MUSCLE program implemented in the Molecular Evolutionary Genetics Analysis version 6 (MEGA 6) [34]. Phylogenetic trees, based on the VP1 nucleotide sequences, were constructed using the maximum-likelihood method in MEGA 6, and the reliability of the generated trees was evaluated by bootstrapping 1000 replicates.

## 3. Results

### 3.1. Particpant Characteristics

A total of 17 infants were enrolled in the study, with four samples collected per infant (a total of *n* = 68 samples were obtained) over a period of six months. Participant demographic information, including age, gender, birth weight, and clinical characteristics including HIV status and vaccination status, was collected as summarised in Table 1.

### 3.2. Enterovirus Species Detected

Based on the results from Genome Detective, which assembles viral reads into contigs by the de novo method and aligns the sequences against a viral reference database for virus identification, four human enterovirus (EV) species were identified, including EV-A, EV-B, EV-C, and EV-D. All 17 infants shed at least a single enterovirus species in their stools at one or more sampling time points specified, and EV-B and EV-C were the most frequently detected species. The virus detection rates for each of the four species, per individual infant, were as follows: EV-A [3/17 (17.6%)], EV-B [16/17 (94.1%)], EV-C [(17/17 (100%)], EV-D [4/17 (23.5%)] (Figure 1; Table 2). 

Regarding the detection rate based on the overall number of samples (n = 68), as expected, EV-C exhibited higher detection frequencies of 55/68 (80.9%), followed by EV-B with 31/68 (45.6%) detections (Table 2). Except for EV-A, all enteroviruses were detected from the first week of birth and throughout the four sampling time points (TPs) (Figure 1; Table 2). Only two species showed 100% detection frequency at a single time point, i.e., EV-B at TP4 and EV-C at TP1 (Figure 1; Table 2).

### 3.3. Identified Enterovirus Serotypes/Genotypes

Using the Enterovirus Genotyping Tool embedded in Genome Detective, detected viruses with sufficient genome coverage of the VP1 protein-coding region, following de novo assembly, could be classified at a lower taxonomic level, and different subtypes/genotypes/serotypes were identified. Overall, nine different EV subtypes were identified in this study. Of the five EV-A species detected, none could be assigned a subtype (Table 2). EV-B species were detected in 31 samples, and five different subtypes were assigned across nine samples. These included echovirus 6 (E6) (n = 1), echovirus 13 (E13) (n = 2), echovirus 15 (E15) (n = 2), echovirus 19 (E19) (n = 3), and coxsackievirus B2 (CVB2) (n = 1). The remaining EV-B species detected (n = 22) could not be assigned subtypes (Table 2).

Enterovirus C was the most frequently detected species (n = 55), and nearly half (n = 27) of these viruses were assigned subtypes. This included poliovirus 1 (PV1) strain Sabin (n = 13), poliovirus 3 (PV3) strain Sabin (n = 10), enterovirus C99 (EV-C99) (n = 3), and coxsackievirus A19 (CVA19) (n = 1), the remaining 28 were unassigned (Table 2). Enterovirus D was only detected in four samples, and as with EV-A, our analysis pipeline was unable to classify them into subtypes (Table 2).

### 3.4. Detection Frequency of Vaccine-Associated Polioviruses 

As described above, the shedding of OPV vaccine-related strains [Sabin poliovirus 1 (PV1) and Sabin poliovirus 3 (PV3)] was observed in this study, and as a result, the rate of viral shedding in these infants’ stools was investigated. The analysis indicates that PV1 strains were predominantly shed in stools from the first week of birth (TP1), with samples from 11 of the 17 infants positive for the virus (Figure 2). The detections dramatically declined to two samples at TP2 (6 weeks). This continued to decline until the virus was completely cleared from the stools at TP4 (24–26 weeks). PV3 was only detected in two samples in the first week of birth before the number of shedders increased to six at TP2. From this point onward, the detection trend is downward, with zero detections at TP4 (24–26 weeks) (Figure 2). 

### 3.5. Detection of Near-Complete Genomes of Non-Polio Enteroviruses

A total of 11 complete/near-complete genomes of NPEV subtypes were identified. These included members of enterovirus B: CVB2 (n = 1), E6 (n = 1), E13 (n = 2), E15 (n = 2), and E19 (n = 1), and members of enterovirus C: CVA19 (n = 1), EV-C99 (n = 3). Descriptions of these genomes are provided in Table 3. The nucleotide sequences determined have been submitted to GenBank.

The complete/near-complete genomes were detected at various time points, but the majority (7/11) were recovered from samples collected between TP4 (24–26 weeks), while three were identified at TP3 (16–20 weeks), and one was detected at TP2 (6–8 weeks) (Table 3).

Regarding the complete genomes, echoviruses had an average sequence length of 7351.5 nucleotide (nt) bases, ranging from 7261 to 7394 nt. Enterovirus C99 had an average length of 7443.3 nt, ranging from 7426 to 7456 nt. Coxsackievirus A19 had a length of 7049 nt. The coverage maps of all 11 genomes mentioned are shown in Appendix A.

### 3.6. Phylogenetic Analysis of Enterovirus C99 Subtypes

To assess the phylogenetic relationships among the three study strains, VRM17C, VRM9D, and VRM17D, and other EV-C99 strains, a phylogenetic tree was generated based on the complete nucleotide sequence of viral capsid VP1 (Figure 3). Based on the generated tree, two strains, i.e., VRM17C and VRM17D, were grouped together and clustered together with a strain from Cameroon isolated in 2014. This was two study strains grouping together, which was not unexpected as these were detected in samples from the same infant, collected at 16–20 weeks and 24–26 weeks, respectively (Figure 3). The third EV-C99 strain from our study (VRM9D) clustered distantly from the other two, alongside strains previously reported in Brazil between 2010 and 2013 (Figure 3). 

## 4. Discussion

Enteroviruses (EVs) are important causative agents of a wide spectrum of illnesses in neonatal age and early infancy. Although efforts to eradicate polioviruses have been successful, infections due to NPEV still contribute to outbreaks of serious diseases such as acute flaccid paralysis and meningitis [36]. This study provides a description of the EVs present in the faeces of infants in their first six months of life. Moreover, this research has proven that metagenomic next-generation sequencing (mNGS) is a promising tool for deciphering the predominant serotypes responsible for EV infections during early childhood. Our findings highlighted that neonates and young infants harbour a diverse population of EVs in their guts, and despite EVs being associated with a range of clinical manifestations, these results further add to the narrative that asymptomatic carriage of EVs is common among the paediatric populations.

Using viral metagenomics, four human EV species (EV-A to EV-D) were detected in faecal specimens of infants who were longitudinally followed from birth until six months of age. Our analysis further revealed the identification of a variety of EV subtypes, mainly from EV-B and EV-C species, whereas none of the viruses from EV-A and EV-D species could be assigned subtypes due to their low genome coverage of the viral capsid protein VP1 coding region. Among the classified subtypes were E6, E13, E15, E19, CVB2, CVA19, PV1, PV3, and EV-C99.

Echoviruses and CVB, which were more common in this cohort, are the most frequent EV subtypes previously reported in other African countries, including Nigeria and Ghana [37,38]. Outside the African continent, they have been reported in China and India, signifying their worldwide distribution [39,40]. Specifically, E6 and E13 are among the most frequently detected EVs worldwide and have been often reported in association with serious diseases in children such as aseptic meningitis and AFP [41,42,43]. E13 has been among the most prevalent types in the Democratic Republic of the Congo and the Central African Republic, while E6 has been found to be prevalent in AFP cases in Nigeria and Cameroon [44,45,46]. In Malawi, E6 and E15 were reported to be the most frequently detected types in Malawian infants at 3.5% and 4.2% detection rates, respectively [47].

Like other EVs, infections due to CVBs are usually moderate; however, they have also been associated with central nervous system diseases, including aseptic meningitis, AFP, and acute encephalitis [48]. There are six types of CVBs (CVB1 to CVB6), and CVB2, which was detected in the current study, accounts for up to 6% of the annual reported cases of EV infections [49]. In serious cases, CVB2 can cause myocarditis, meningoencephalitis, and hand, foot, and mouth disease (HFMD) [50,51].

Coxsackievirus A19, identified in this study, is a rare EV type belonging to the species EV-C [52], and it has been associated with illnesses such as gastroenteritis and acute enteritis [53,54]. Consistent with our study, a previous faecal virome study from the Amazon detected CVA19 in stool samples of healthy children [55]. Recently, CVA19 was identified in a stool sample of a patient with HFMD in China [56]. Taken together, these findings highlight the abundance of CVA19 in the human gastrointestinal tract, and in support of this, evidence from the literature showed that more than 50% of the sequences of CVA19 in the GenBank database were isolated from human faecal samples [56].

Poliovirus Sabin vaccine strains were the most predominantly detected serotypes in this study, belonging to EV-C species. Sabin PV1 was more prevalent during the first week of birth, corresponding with the first dose of OPV, and steadily declined until it was cleared by 24 weeks of age. Similarly, PV3 peaked at six weeks, corresponding to the second dose of OPV, and was also cleared by 24 weeks. The high rate of Sabin PVs is attributable to the shedding of oral vaccines administered at birth and six weeks against PV infection.

Enterovirus C99 (EV-C99) is a relatively new EV subtype within the species Enterovirus C. It was first identified in Bangladesh in 2000 [57] and subsequently reported worldwide, including in non-human primates [58,59,60]. In Africa, EV-C99 was previously identified in individuals with AFP and healthy children from Cameroon, Ivory Coast, and Nigeria [46,61,62]. A cross-sectional study involving a cohort of Malawian infants revealed that EV-C99 was the second most frequently detected subtype (~10%) within EV-C species, after CVA13 [47].

As of 2019, only 15 complete genomes of EV-C99 were available in GenBank [63]. In the current study, three complete genomes of EV-C99 were detected in two infants, adding to the collection of EV sequences in current genome databases. Phylogenetic analysis of these strains based on the VP1 gene showed that two of the strains, i.e., VRM17C and VRM17D, detected in stool samples from the same infant, were closely related to strains isolated in Cameroon using viral metagenomics [64]. The third EV-C99 clustered together with strains from Brazil, the closest being strain BRA/TO-16 detected in a sample collected in 2013 from a female child having symptoms of gastroenteritis such as diarrhoea and vomiting [63]. Together with our findings, this demonstrates the global distribution of EV-C99 as well as its role both in gastroenteritis and in asymptomatic infections. 

Although EVs primarily spread from person to person via the faecal-oral and respiratory routes, numerous other modes of transmission have been well documented. Previous studies have reported that EV infections can be acquired vertically during delivery, often through contact with maternal blood, faeces, or genital secretions [65]. Recent data also highlighted the possibility of horizontal transmission of EVs through contact with family members after delivery. Specifically, a study conducted in Japan on neonates and young infants identified EVs in more than 90% of siblings, many of whom were asymptomatic [66]. 

On the other hand, there have been conflicting reports on the potential role of breastfeeding in the transmission of EVs. While an earlier study by Sadeharju and colleagues (2007) demonstrated that the duration of breastfeeding may play a protective role against EV infection [67], other authors suggested that breastfeeding could be a source of EV transmission [68,69]. In a study where a total of 150 infants were prospectively followed from birth up to one year of age, maternal breastmilk and blood samples were screened for the presence of EV antibodies and RNA [67]. Fewer EV infections were observed in infants who were exclusively breastfed for more than two weeks, compared to those breastfed for two weeks or less. Subsequently, an indirect correlation was seen between the levels of maternal antibodies in breast milk and infection rates, associating high antibody levels with a decrease in infection frequencies. In this study, no EVs were reported in breastmilk [67]. 

Conversely, EVs were detected in the breast milk of two symptomatic mothers of neonates diagnosed with severe hepatitis and meningitis [68], while in another study, EVs were present in breastmilk obtained from a parent of an infant with sepsis [69]. It was, however, not known whether breastmilk, as a potential source of EV transmission, contributed to the disease severity. It would therefore be of great significance for future research to consider case-control approaches when investigating EV infections in neonatal health conditions such as these.

In our study, the comparison between different breastfeeding durations was not practical as all infants were breastfeeding for the entire duration of the study, i.e., six months or more, and none of the infants presented with any EV-associated disease. However, how EVs in the current cohort were acquired as early as less than seven days old remains to be interrogated. 

Given the extreme variability of predominant EV subtypes across geographical locations and the demonstrated utility of mNGS, intensified genomic surveillance is necessary to elucidate the diversity of EVs in early childhood. This would improve our understanding of their potential contributions to disease burden and enable a prompt response in cases of outbreaks. Establishing such an effective NPEV pathogen surveillance system on the African continent, where there is a high burden of infectious diseases, would provide crucial information on the circulation of known and rare EV subtypes locally and globally. 

Although samples from our study were collected more recently, providing more relevant information on the currently circulating EV types, the major limitation is the shorter sampling period. Samples collected over longer periods will provide a more comprehensive image of the diversity and persistence of EVs circulating in paediatric populations.

## 5. Conclusions

The genus *Enterovirus* in the family *Picornaviridae* encompasses a large group of NPEVs that have been reported in a wide spectrum of illnesses. The current study adds to our understanding of the types of EVs that healthy South African infants are constantly exposed to. Research on such viruses, which have the potential to cause severe life-threatening health complications, has laid important foundations for studies seeking to characterise viral pathogens colonising the gut of newborn babies. Although poliovirus is on the verge of complete eradication, more efforts must be intensified to understand the role of NPEV as a non-polio aetiologic agent of neonatal diseases. In conclusion, the current data underlines the importance of developing long-term measures to monitor NPEV infections in early life.

## Figures and Tables

**Figure 1 microorganisms-12-00920-f001:**
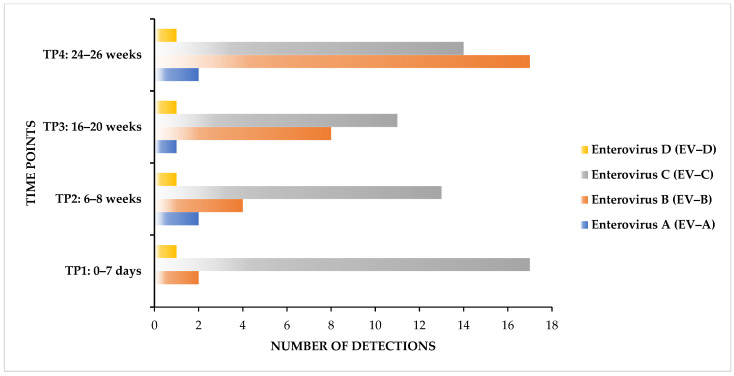
Enterovirus species A–D detections in the stool of infants at each of the four time points (TP).

**Figure 2 microorganisms-12-00920-f002:**
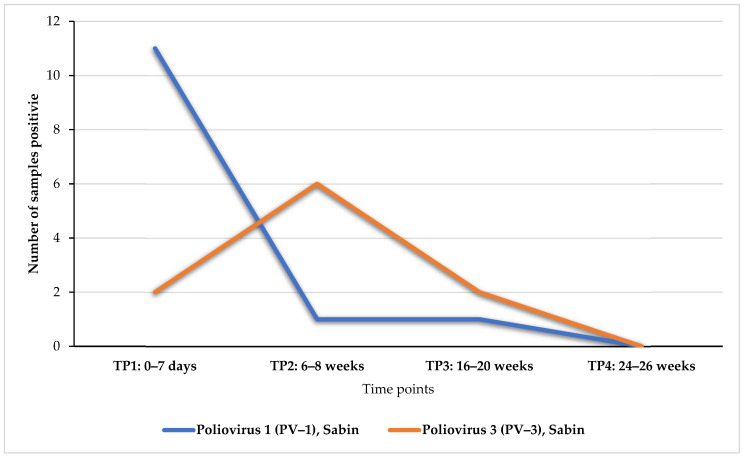
Rate of faecal shedding of poliovirus-related vaccine strains.

**Figure 3 microorganisms-12-00920-f003:**
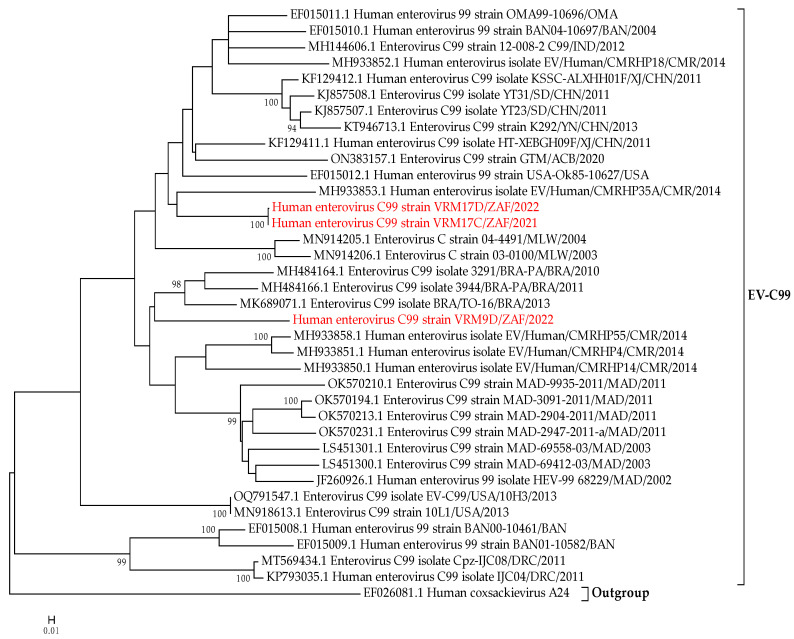
Maximum-likelihood phylogenetic tree based on the complete VP1 nucleotide sequence of enterovirus C99 strains (EV-C99). The evolutionary history was inferred by using the Maximum Likelihood method on MEGA version 6 [34], based on the General Time Reversible (GTR) model [35]. Values on the nodes represent the bootstrap support of the node (1000 bootstrap replicates). A discrete Gamma distribution was used to model evolutionary rate differences among sites (five categories (+G, parameter = 1.1867)). The rate variation model allowed for some sites to be evolutionarily invariable ([+I], 49.4359% sites).

**Table 1 microorganisms-12-00920-t001:** Clinical and demographic information of study participants.

**Total number of infants**	17
**Males**	5 (29.4%)
**Females**	12 (70.6%)
**Average gestational period**	38.5 weeks
**Average birth weight**	2875 g
**Vaginal delivery**	14 (82.4%)
**Caesarean section**	3 (17.6%)
**Breastfeeding**	15 (88.2%)
**Mixed feeding**	2 (11.8%)
**HIV exposed**	5 (29.4%)
**HIV non-exposed**	11 (64.7%)
**Unknown HIV status**	1 (5.9%)
**OPV vaccination status**	All were vaccinated at birth and at six weeks
**Symptomatic diarrhoea**	None
**Other illnesses**	None

**Table 2 microorganisms-12-00920-t002:** Summary of enterovirus species and subtypes detected in infants at different time points. Enteroviruses A to D and their respective subtypes were reported.

Species	Subspecies/Subtype	Number of Infants with the Detected Virus	Total Number of Samples with the Detected Virus	Number of Samples with the Detected Virus per Time Point (TP)			
				TP1 (0–7 Days)	TP2 (6–8 Weeks)	TP3 (16–20 Weeks)	TP4 (24–26 Weeks)
**Enterovirus A (EV-A)**		**3/17 (17.6%)**	**5**		2	1	2
	Unassigned EV-A		**5**		2	1	2
**Enterovirus B (EV-B)**		**16/17 (94** **.** **1%)**	**31**	2	4	8	17
	Echovirus 6 (E6)		**1**				1
	Echovirus 13 (E13)		**2**		1		1
	Echovirus 15 (E15)		**2**			1	1
	Echovirus 19 (E19)		**3**				3
	Coxsackievirus B2 (CVB2)		**1**			1	
	Unassigned EV-B		**22**	2	3	6	11
**Enterovirus C (EV-C)**		**17/17 (100%)**	**55**	17	13	11	14
	Poliovirus 1 (PV1), Sabin		**13**	11	1	1	
	Poliovirus 3 (PV3), Sabin		**10**	2	6	2	
	Enterovirus C99 (EV-C99)		**3**			1	2
	Coxsackievirus A19 (CVA19)		**1**				1
	Unassigned EV-C		**28**	4	6	7	11
**Enterovirus D (EV-D)**		**4/17 (23.5%)**	**4**	1	1	1	1
	Unassigned EV-D		**4**	1	1	1	1

**Table 3 microorganisms-12-00920-t003:** Information on coverage and sequence length of the 11 NPEV complete and near-complete genomes recovered in the study, as well as GenBank accession numbers.

Species	Subtype		Coding Sequence Coverage	Sequence Length (NT)	Sample ID	Time Point (Age Range)	GenBank Accession Numbers
**Enterovirus B**	Coxsackievirus B2	CVB2	99.4%	7120	VRM2C	TP3 (16–20 weeks)	PP711772
	Echovirus 6	E6	99.4%	7546	VRM10D	TP4 (24–26 weeks)	PP711773
	Echovirus 13	E13	93.4%	7281	VRM15B	TP2 (6–8 weeks)	PP711774
		E13	100%	7261	VRM17D	TP4 (24–26 weeks)	PP711775
	Echovirus 15	E15	100%	7362	VRM15C	TP3 (16–20 weeks)	PP711776
		E15	100%	7394	VRM15D	TP4 (24–26 weeks)	PP711777
	Echovirus 19	E19	100%	7389	VRM4D	TP4 (24–26 weeks)	PP711778
**Enterovirus C**	Coxsackievirus A19	CVA19	100%	7049	VRM1D	TP4 (24–26 weeks)	PP711771
	Enterovirus C99	EV-C99	100%	7426	VRM17C	TP3 (16–20 weeks)	PP711781
		EV-C99	100%	7456	VRM9D	TP4 (24–26 weeks)	PP711779
		EV-C99	100%	7448	VRM17D	TP4 (24–26 weeks)	PP711780

## Data Availability

The data presented in this study is available upon request.

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
