# Peer review of "High Detection Frequency of Vaccine-Associated Polioviruses and Non-Polio Enteroviruses in the Stools of Asymptomatic Infants from the Free State Province, South Africa"

_microorganisms, 2024, doi:10.3390/microorganisms12050920_

Round 1
Reviewer 1 Report
Comments and Suggestions for Authors
In this study, the authors identified human enterovirus from stool samples of healthy newborns in South Africa. The authors analyzed enteroviral genomes in the samples. The results would provide information on enteroviruses circulating in Africa.
Specific comments:
- Abstract: There are no PV-2 isolates in this study.
- Table 2: Mahoney, Sabin 1, Leon, and Sabin 3 are different strains. Please clarify the strain names.
- The selection criteria for the newborns should be clarified. Did they have some clinical symptoms?
- Table 2: Status of genome analysis (nearly complete sequence, VP1 only, etc.), patient number for the identified EV, GenBank accession should be included in Table 2 for each identified EV.
- Table 2: The definition of positive is unclear.
- Nucleotide sequences identified in this study should be deposited to GenBank and the accession numbers should be available.
- Genome coverage maps may be moved to supplementary data.
- L112: I?
Author Response
Reviewer 1
Specific comments:
1. Abstract: There are no PV-2 isolates in this study.
This was a typo and has been rectified to PV-3.
2. Table 2: Mahoney, Sabin 1, Leon, and Sabin 3 are different strains. Please clarify the strain names.
Both PV1 and PV3 detected were Sabin vaccine strains and this has been corrected in Table 2.
3. The selection criteria for the newborns should be clarified. Did they have some clinical symptoms?
The selection criteria included all newborns with and/or without clinical symptoms, however, newborns with congenital abnormalities that required hospitalizations were excluded, and this has been described in the main manuscript document.
4. Table 2: Status of genome analysis (nearly complete sequence, VP1 only, etc.), patient number for the identified EV, GenBank accession should be included in Table 2 for each identified EV.
The GenBank accessions and sample IDs have been included as per reviewer’s recommendation. However, they best fit in Table 3 which describes all the nearly complete genomes.
5. Table 2: The definition of positive is unclear.
This has been changed in Table 3.
6. Nucleotide sequences identified in this study should be deposited to GenBank and the accession numbers should be available.
The sequences have been deposited in GenBank and the accession numbers are provided in Table 3.
7. Genome coverage maps may be moved to supplementary data.
The genome coverage maps have been moved to supplementary material as advised by the reviewer.
8. L112: I?
The typo error has been corrected.
Reviewer 2 Report
Comments and Suggestions for Authors
ABSTRACT
Lines 21, 25:
“ …poliovirus subtypes (PV-1 and PV-2) were identified”
“Sabin PV-1 and PV-2 strains were two …”
Apparently, this is a mistake. There is no mention of poliovirus type 2 in the Results.
INTRODUCTION
Lines 57-59: “Newborns are more vulnerable to EV infections, with an incidence of seven cases per 1000 newborns (Jenista et al., 1984; Chuang and Huang, 2019; 58 Moliner-Calderón, 2023).” –
These references are not in the reference list - Chuang and Huang, 2019; Moliner-Calderón, 2023.
The link (Jenista et al., 1984) is repeated twice under the numbers 25 and 27.
Please check that this link (Jenista et al., 1984) is correct. The link is very old, the situation may change in 40 years.
Line 63: reference Chuang and Huand, 2019 are not in the reference list. Please. check your spelling – Huang or Huand?
Line 64: “… PV-1, 64 PV-2, and PV-3” - Here and further, please adhere to the abbreviated name of viruses according recommendations of Simmonds et al. Recommendations for the nomenclature of enteroviruses and rhinoviruses. Arch Virol. 2020 Mar;165(3):793-797. doi: 10.1007/s00705-019-04520-6.
Lines 73-74: “As a result of effective immunisation and intensive surveillance, by 2015, there were zero cases of polio.” - It is very important to add that we are talking about wild poliovirus. In 2023, vaccine-derived polioviruses (VDPV) continued to circulate in the African Region (please see WHO website).
Line 81: “…oral polio vaccine (tOPV) …” - This abbreviation has already occurred (line 78).
Line 90: “Nevertheless, many of the studies on enteroviruses in South Africa have focused on AFP cases.” - Please provide references.
Lines 90-93 - The logic of moving from a surveillance study of AFP cases to a short-time study of newborn stool samples is not very clear. Are there other types of poliovirus surveillance in South Africa, such as wastewater surveys? Is there surveillance of non-polio enterovirus infections? Please describe the purpose of the study more clearly. Why this particular group of children was chosen especially considering that the children were immunized with live oral poliovirus vaccine?
MATERIALS AND METHODS
Line 107: Please note the link in the form of a number [3].
RESULTS
Lines 160-161: “including enterovirus A 160 (EV-A), enterovirus B (EV-B), enterovirus C (EV-C), and enterovirus D (EV-D).” - These abbreviations are more appropriately introduced in the Introduction section.
Lines 187-188: “These included poliovirus 1 (PV-1) strain Mahoney/Sabin (n=13), poliovirus 3 (PV-3) strain Leon/Sabin (n=10), enterovirus C99.” - What strains of poliovirus were detected? Mahoney and Leon are wild prototype polioviruses and cannot be present in bOPV that children have been mmunized. And they do not circulate in nature. Please clarify.
3.5. Detection of near-complete genomes of non-polio enteroviruses
Lines 14-18: “A total of 11 complete/near complete genomes of NPEV subtypes were identified. These included members of enterovirus B [coxsackievirus B2 (CV-B2) (n=1), echovirus 6 (E-6) (n=1), echovirus 13 (E-13) (n=2), echovirus 15 (E-15) (n=2), echovirus 19 (n=1)] and 17 members of enterovirus C [coxsackievirus A19 (n=1), enterovirus C99 (EV-C99) (n=3)].” - Please do not use square brackets. Abbreviated names of viruses have already been given.
DISCUSSION
Lines 67-76: “Enteroviruses (EVs) are important causative agents of a wide spectrum of illnesses in neonatal age and early infancy. Although efforts to eradicate polioviruses have been a success, infections due to NPEV still contribute to outbreaks of serious diseases such as acute flaccid paralysis and meningitis (Suresh et al., 2019). This study provides a description of the EVs present in faeces of infants in their first six months of life. Moreover, this research has proven that metagenomic next generation sequencing (mNGS) is a promising tool in deciphering the predominant serotypes responsible for EV infections during early childhood. Our findings highlighted that neonates and young infants harbour a diverse population of EVs in their guts, and despite EVs being associated with a range of clinical manifestations, these results further add to the narrative that asymptomatic carriage of EVs is common among the paediatric populations.” - It is difficult to agree with this statement. The study group was a group of healthy children without clinical manifestations of enterovirus infection. The results obtained indicate a gradual “colonization” of the intestine with microflora, which includes non-polio enteroviruses. Please state more precisely.
Lines 83-84 - Virus abbreviations are repeated again.
Line 85: “ … (PV-1, PV-2, PV-3) - Please check what's wrong with poliovirus type 2 - it is not part of bOPV, all over the world PV2 is preserved under containment conditions.
Line 112 –“ … and I support of this …” - Please check the style.
I believe that the text of the Discussion could be shortened, especially in the part devoted to the effects of breastfeeding. Please explain the benefits of testing stool samples from newborn infants before routine surveillance for enterovirus infections, given that most non-polio enteroviruses do not cause clinical manifestations.
CONCLUSION
Line 192-194: “Although poliovirus is on the verge of complete eradication, more efforts must be intensified in understanding the role of NPEV as non-polio aetiologic agents.” - Please clarify this – non-polio aetiological agents of what?

Author Response
ABSTRACT
Lines 21, 25:
“ …poliovirus subtypes (PV-1 and PV-2) were identified”
“Sabin PV-1 and PV-2 strains were two …”
Apparently, this is a mistake. There is no mention of poliovirus type 2 in the Results.
This error has been rectified accordingly.
INTRODUCTION
Lines 57-59: “Newborns are more vulnerable to EV infections, with an incidence of seven cases per 1000 newborns (Jenista et al., 1984; Chuang and Huang, 2019; 58 Moliner-Calderón, 2023).” –
These references are not in the reference list - Chuang and Huang, 2019; Moliner-Calderón, 2023.
The link (Jenista et al., 1984) is repeated twice under the numbers 25 and 27.
Please check that this link (Jenista et al., 1984) is correct. The link is very old, the situation may change in 40 years.
These oversights have been attended to and addressed.
Line 63: reference Chuang and Huand, 2019 are not in the reference list. Please. check your spelling – Huang or Huand?
This was a typo and it has been attended to.
Line 64: “… PV-1, 64 PV-2, and PV-3” - Here and further, please adhere to the abbreviated name of viruses according recommendations of Simmonds et al. Recommendations for the nomenclature of enteroviruses and rhinoviruses. Arch Virol. 2020 Mar;165(3):793-797. doi: 10.1007/s00705-019-04520-6.
All the abbreviations have been changed according to the Simmonds et al recommendations.
Lines 73-74: “As a result of effective immunisation and intensive surveillance, by 2015, there were zero cases of polio.” - It is very important to add that we are talking about wild poliovirus. In 2023, vaccine-derived polioviruses (VDPV) continued to circulate in the African Region (please see WHO website).
Noted and revised as recommended.
Line 81: “…oral polio vaccine (tOPV) …” - This abbreviation has already occurred (line 78).
This has been addressed as advised.
Line 90: “Nevertheless, many of the studies on enteroviruses in South Africa have focused on AFP cases.” - Please provide references.
The reference has been provided.
Lines 90-93 - The logic of moving from a surveillance study of AFP cases to a short-time study of newborn stool samples is not very clear. Are there other types of poliovirus surveillance in South Africa, such as wastewater surveys? Is there surveillance of non-polio enterovirus infections? Please describe the purpose of the study more clearly. Why this particular group of children was chosen especially considering that the children were immunized with live oral poliovirus vaccine?
The current study describes the enterovirus strains that were identified in asymptomatic newborns as part of gut viral metagenomic study in the Free State. What is reported here is a subset of the data from the main study focusing on the EV genomes which were analysed further to gain insight into the diversity and prevalence of these EVs.
MATERIALS AND METHODS
Line 107: Please note the link in the form of a number [3].
The link has been replaced with the correct in text reference.
RESULTS
Lines 160-161: “including enterovirus A 160 (EV-A), enterovirus B (EV-B), enterovirus C (EV-C), and enterovirus D (EV-D).” - These abbreviations are more appropriately introduced in the Introduction section.
Revised as recommended.
Lines 187-188: “These included poliovirus 1 (PV-1) strain Mahoney/Sabin (n=13), poliovirus 3 (PV-3) strain Leon/Sabin (n=10), enterovirus C99.” - What strains of poliovirus were detected? Mahoney and Leon are wild prototype polioviruses and cannot be present in bOPV that children have been mmunized. And they do not circulate in nature. Please clarify.
The wildtype strain names have been removed as these were vaccine strains.
3.5. Detection of near-complete genomes of non-polio enteroviruses
Lines 14-18: “A total of 11 complete/near complete genomes of NPEV subtypes were identified. These included members of enterovirus B [coxsackievirus B2 (CV-B2) (n=1), echovirus 6 (E-6) (n=1), echovirus 13 (E-13) (n=2), echovirus 15 (E-15) (n=2), echovirus 19 (n=1)] and 17 members of enterovirus C [coxsackievirus A19 (n=1), enterovirus C99 (EV-C99) (n=3)].” - Please do not use square brackets. Abbreviated names of viruses have already been given.
The square brackets and full names have been removed, retaining the abbreviated names.
DISCUSSION
Lines 67-76: “Enteroviruses (EVs) are important causative agents of a wide spectrum of illnesses in neonatal age and early infancy. Although efforts to eradicate polioviruses have been a success, infections due to NPEV still contribute to outbreaks of serious diseases such as acute flaccid paralysis and meningitis (Suresh et al., 2019). This study provides a description of the EVs present in faeces of infants in their first six months of life. Moreover, this research has proven that metagenomic next generation sequencing (mNGS) is a promising tool in deciphering the predominant serotypes responsible for EV infections during early childhood. Our findings highlighted that neonates and young infants harbour a diverse population of EVs in their guts, and despite EVs being associated with a range of clinical manifestations, these results further add to the narrative that asymptomatic carriage of EVs is common among the paediatric populations.” - It is difficult to agree with this statement. The study group was a group of healthy children without clinical manifestations of enterovirus infection. The results obtained indicate a gradual “colonization” of the intestine with microflora, which includes non-polio enteroviruses. Please state more precisely.
We are in agreement with the reviewer, and the statement explains that despite EVs being associated with some clinical manifestations, asymptomatic infection is also very common.
Lines 83-84 - Virus abbreviations are repeated again.
Corrected.
Line 85: “ … (PV-1, PV-2, PV-3) - Please check what's wrong with poliovirus type 2 - it is not part of bOPV, all over the world PV2 is preserved under containment conditions.
This has been revised.
Line 112 –“ … and I support of this …” - Please check the style.
The typo has been corrected.
I believe that the text of the Discussion could be shortened, especially in the part devoted to the effects of breastfeeding. Please explain the benefits of testing stool samples from newborn infants before routine surveillance for enterovirus infections, given that most non-polio enteroviruses do not cause clinical manifestations.
Thank you for the important point raised here. However, we believe that although it is reported that most non-polio enteroviruses do not cause clinical manifestations it is also possible that their roles in clinical illnesses have not been established as yet. Therefore testing stool samples especially in the early stages of life will be clinically beneficial as this can provide insights not only on the diversity of enterovirus serotypes but may also reveal associations with demographic variables such as socio-economic status, feeding practices, mode of delivery, environmental factors, ethnicity, living conditions, HIV exposure among others.
CONCLUSION
Line 192-194: “Although poliovirus is on the verge of complete eradication, more efforts must be intensified in understanding the role of NPEV as non-polio aetiologic agents.” - Please clarify this – non-polio aetiological agents of what?
NPEVs have been reported in a wide spectrum of illnesses however their role in neonatal diseases has not been fully investigated, thus the statement refers to aetiologic agent of illnesses in newborns.